palaeontology

laser-stimulated fluorescence, ultraviolet, techniques, Jurassic, Solnhofen Limestones

**Author for correspondence:**
Luke A. Barlow
e-mail: barlowpalaeo@gmail.com

†Lead Contact.

# Laser-stimulated fluorescence reveals unseen details in fossils from the Upper Jurassic Solnhofen Limestones

Luke A. Barlow[1,2,†], Michael Pittman[1], Anthony Butcher[2], David M. Martill[2] and Thomas G. Kaye[3]

[1]Vertebrate Palaeontology Laboratory, Department of Earth Sciences, The University of Hong Kong, Pokfulam, Hong Kong
[2]School of the Environment, Geography and Geosciences, University of Portsmouth, Burnaby Building, Portsmouth, UK
[3]Foundation for Scientific Advancement, Sierra Vista, AZ 85650, USA

LAB, 0000-0002-0867-649X; MP, 0000-0002-6149-3078

Laser-stimulated fluorescence (LSF) has seen increased use in palaeontological investigations in recent years. The method uses the high flux of laser light of visible wavelengths to reveal details sometimes missed by traditional long-wave ultraviolet (UV) methods using a lamp. In this study, we compare the results of LSF with UV-A-generated fluorescence on a range of fossils from the Upper Jurassic Solnhofen Limestone Konservat-Lagerstätte of Bavaria, Germany. The methodology follows previous protocols of LSF with modifications made to enhance laser beam intensity, namely keeping the laser at a constant distance from the specimen, using a camera track. Our experiments show that along with making surface details more vivid than UV-A or revealing them for the first time, LSF has the additional value of revealing shallow subsurface specimen detail. Fossil decapods from the Solnhofen Limestone reveal full body outlines, even under the matrix, along with details of segmentation within the appendages such as limbs and antennae. The results indicate that LSF can be used on invertebrate fossils along with vertebrates and may often surpass the information provided by traditional UV methods.

## 1. Introduction

Laser-stimulated fluorescence (LSF) is a non-destructive imaging method involving visible wavelengths that were recently

introduced to palaeontological studies and have various applications [1]. This includes the potential of LSF to reveal fossils lying a small distance below the matrix surface and its integration into an automated microvertebrate sorting machine. The machine uses a bowl feeder where samples are excited by a laser and those with a fluorescence index above certain values indicating a fossil is blown into a separate dish, therefore reducing the time spent on manual sorting [1] (figures 8 and 10). Recently, LSF has been implemented as part of an autonomous drone system that seeks out fossils at night, with tests including the recovery of vertebrate fossils from the badlands of Wyoming [2]. The basic methodology is simple as high-quality images of fluorescing fossils can be produced rapidly with a 30-second exposure in a darkened room. The green laser light wavelength of 532 nm has been previously employed [1], but more recently available high power blue/violet lasers have superseded the use of green wavelengths. Blue/violet lasers produce similar fluorescent colours to UV-A but the laser produces a stronger fluorescence signal [1]. As reported by several studies [3–10], LSF continues to reveal new and exciting details in well-preserved fossil assemblages (Konservat-Lagerstätten) as seen in the Yanliao [3–5,11,12], Jehol [8,10,13–15] and Las Hoyas Lagerstätten [16] of northeastern China and Spain, respectively.

Here we conduct LSF and UV-A imaging on fossil specimens from the Solnhofen Lagerstätten of southern Germany and make imaging recommendations based on our results. These specimens fluoresce brightly due to high levels of phosphate present in the skeletons and the surrounding environment [17]. The Solnhofen Limestones are famous for their well-bedded, ultrafine-grained lithographic limestones (often called Plattenkalk and referred to as such herein) that formed in calm basins of a closed lagoonal system on the northern margin of the Tethys Ocean [18,19]. These provide a rare window into the fossil record of the Jurassic [20]. The high evaporation rates in these lagoons resulted in a stratified water column with anoxic bottom waters largely devoid of macro-organisms [18]. Occasional mixing through storms brought the toxic water to the aerated surface zone leading to the mass mortality of free-swimming pelagic organisms [20]. These organisms often became exceptionally well preserved due to the inhospitable conditions above the sediment–water interface which excluded scavenging organisms, through microbial film formation and/or rapid burial [21]. The excellent preservation is rare and, therefore, the fossils are often too valuable to undergo destructive analysis.

The list of non-destructive techniques available to palaeontologists is increasing and a wide range has been used to study Solnhofen fossils. X-rays were first used for palaeontological studies in 1896, and were widely implemented from 1934 onwards by Lehmann on the Solnhofen Limestone [22]. Composite imaging through focus stacking [23], three-dimensional computer modelling using CT scanning [24] and red-cyan anaglyphs have also been used [25], providing valuable alternatives to more destructive techniques, especially on small and delicate specimens [26,27]. Fluorescence does not work on all specimens, for instance, Sabroux *et al.* tried UV-A (348 nm) and blue-green light (460 nm) on fossil pycnogonids without discernible fluorescence [28]. Green-orange light-based fluorescence (546 nm) has also been used on material that remains dark under UV [29]. Synchrotron Rapid Scanning X-ray Fluorescence (SRS-XRF) had produced chemical images that map the elements within the plumage of the iconic Solnhofen bird *Archaeopteryx* [30].

Since the introduction of UV fluorescence in analysing fossils from the Solnhofen limestones in the early twentieth century [31], the technique has been increasingly used in the search for (*sensu* Croft *et al.* [32]) and analyses of its exceptionally preserved fossils. Studies of Solnhofen fossils using UV include those of decapod crustaceans [33–35], ammonites [36,37], fish [38,39], pterosaurs [40,41] and dinosaurs [42–46] among others. In recent years, laser light of 405–532 nm wavelengths has been applied to fossils to induce fluorescence and reveal clearer and/or otherwise unseen details [1,47], especially on soft-tissue bearing fossils pertinent to the study of avian and flight origins and integument evolution [1,3–5,8–13,15,16,48,49]. LSF has also been used to study the first discovered fossil feather previously attributed to *Archaeopteryx lithographica* [50] of the Solnhofen Limestones, which is now thought to belong to a different theropod [48]. LSF has also been used to report feather moulting in the Thermopolis specimen of *Archaeopteryx* [49]. The historical convention of using UV to study Solnhofen fossils provides an ideal opportunity to compare images of Solnhofen fossils under UV and LSF to evaluate the efficacy of using LSF to study fossils from this iconic Konservat-Lagerstätte. This study aims to provide a comparative framework to inform fluorescent imaging best practices for working on flat fossils from this and other Lagerstätten [51,52].

## 2. Material and methods

### 2.1. Studied material

The specimens used in this study were collected during a series of field visits to Daiting, Eichstätt, Ettling and Solnhofen in Bavaria, Southern Germany and are accessioned in the collection of the

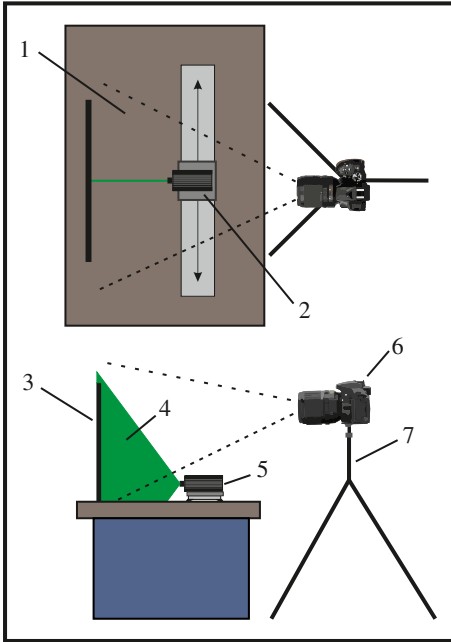

**Figure 1.** A simplified diagram of the camera and specimen table set-up used to perform track based LSF imaging. 1, the image area of the DSLR camera; 2, carriage of the camera track holding the laser module; 3, the imaged specimen; 4, laser illumination using a line generator; 5, 85 mW 532 nm laser module; 6, DSLR 5300; 7, tripod for long exposure photography. Not to scale.

School of the Environment, Geography and Geosciences (SEGG), at the University of Portsmouth (UOP-PAL-SOL-001–013). Additional specimens from the Staatliche naturwissenschaftliche Sammlungen Bayerns, Bayerische Staatssammlung für Paläontologie und Geologie (SNSB-BSPG 1937 I 27, figure 9; SNSB-BSPG 1935 I 24, electronic supplementary material, figure S5; SNSB-BSPG AS I 745a, electronic supplementary material, figure S6; SNSB-BSPG 1977 XIX 1, electronic supplementary material, figure S7) were studied at the Bavarian State Collections of Palaeontology and Geology, Munich. Cephalopods occur frequently in the Solnhofen Limestones as ammonites, belemnites and teuthoids, and are sometimes exceptionally well preserved. In this study, teuthoids are preserved with a full gladius, and ammonites vary from dissolved specimen outlines with *in situ* aptychi to individual aptychi (figure 2; electronic supplementary material, figures S1–S4). Decapod crustaceans (UOP-PAL-SOL-006–009) with original exoskeletal material can be studied as articulated specimens or isolated elements (figures 3–7). Vertebrates are a rare but famous component in the Solnhofen Limestone and are studied here, in the form of fish and reptiles (figures 8 and 9 and electronic supplementary material, figures S5–S10). The specimens used are largely unprepared, lending themselves to studies under fluorescence, and represent the major groups found in the Solnhofen Limestones.

## 2.2. Track-based laser-stimulated fluorescence

The method is a variant of LSF from Kaye *et al.* [1]; it uses an MGL-III-532–1 approximately 300 mW green diode-pumped solid-state (DPSS) laser with a PSU-III-LCD power supply with a set output of 85 mW. Main modifications to the original method include mounting the laser module onto a track instead of a tripod and having the specimens set-up vertically. These two aspects contrast with previous LSF publications and other epifluorescence studies where the specimen lays flat, and the camera is mounted above using a tripod or copy stand. The camera track allowed the 532 nm green laser to scan across the entirety of specimens UOP-PAL-SOL-001–013. The laser is normal to the fossil as it moves through the $x$-axis, maintaining constant beam intensity (figure 1). See figures 2–8 for corresponding method.

## 2.3. Tripod-based laser-stimulated fluorescence

The original method of LSF from Kaye *et al.* [1] is included below and was used on specimens (SNSB-BSPG 1937 I 27; SNSB-BSPG 1935 I 24; SNSB-BSPG AS I 745a; SNSB-BSPG 1977 XIX 1) housed in the

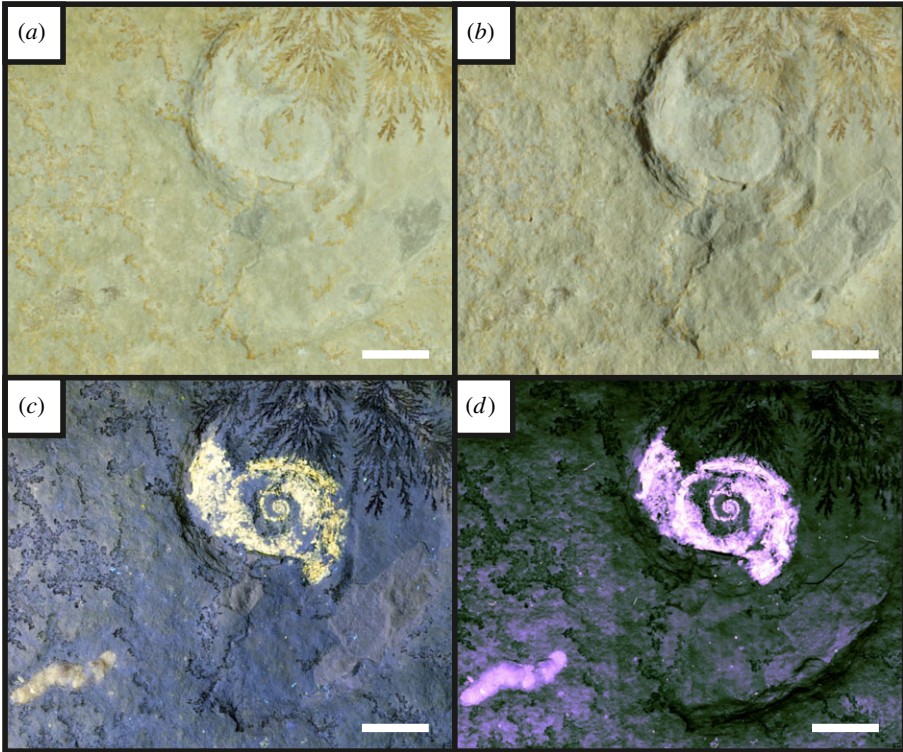

**Figure 2.** The oppeliid ammonite *Neochetoceras* sp. [53] with some phragmacone fluorescence and the putative ammonite coprolite *Lumbricaria* attributed to UOP-PAL-SOL–004. (*a*) direct white light recording a slight colour difference around the phragmacone and very little evidence of the coprolite; (*b*) oblique white light; (*c*) fluorescence of the coprolite, siphuncle and phragmacone under UV-A light; (*d*) LSF displaying full fluorescence of the phragmacone, increasing the observed contrast. Note the brighter siphuncle under 532 nm green laser light in (*d*), allowing for a greater distinction than under UV-A light seen in C. Scale bar = 10 mm.

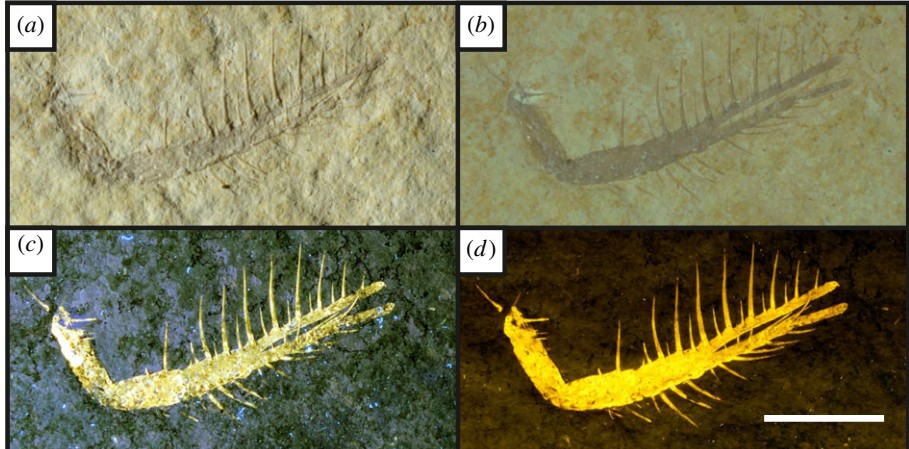

**Figure 3.** An isolated pereiopod of the fossil prawn *Aeger* sp. [54] under different lighting conditions (*a–d*) (UOP-PAL-SOL–006). (*a*), oblique white light image, casting shadows on the slight relief present; (*b*), direct white light; (*c*), The same specimen under 365 nm UV-A fluorescence, highlighting the specimen from the background and recovering some unseen setae that appear broken; (*d*), Laser-stimulated fluorescence image of the specimen under 532 nm green laser displaying an improvement over the UV-A image seen in (*c*) with the complete leg revealed along with none of the gaps present under UV-A. Scale bar = 10 mm.

Bavarian State Collections of Palaeontology and Geology, Munich. A 1 W 405 nm violet laser contained in a panning mount was mounted on a tripod. This was used to induce fluorescence and long-exposure LSF photographs were taken using a Nikon D810 DSLR fitted with a 425 nm longpass blocking filter. See figure 9 for corresponding method.

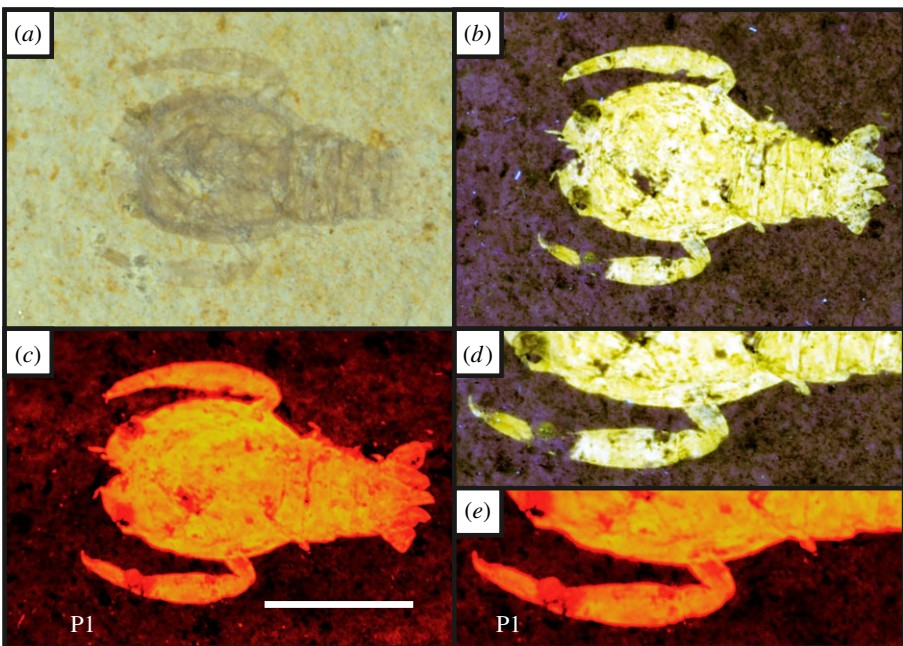

**Figure 4.** A juvenile of the eryonid crustacean *Knebelia* sp. [55] under white light (*a*), 365 nm UV-A (*b*) and LSF (*c*) revealing the full outline of the animal under a 532 nm green laser (UOP-PAL-SOL–007). (*d*) and (*e*) represent comparisons of the fluorescent techniques on the first pereiopod (P1). Note the missing section under UV-A light is revealed through the subsurface illumination of LSF. Scale bar = 10 mm. (*d* and *e*) magnified ×1.5.

## 2.4. UV fluorescence

The method of ultraviolet-induced fluorescence consisted of a 365 nm lamp (UV-A) [40] and followed the procedure used in Tischlinger & Frey [60] with the specimen illuminated as closely as possible. Exposures of more than 10 s did not reveal additional information.

## 2.5. Photography

Photographs for track-based LSF and tripod-based LSF were taken in a blacked-out room to avoid white light contamination.

For tripod-based LSF, a Nikon D810 DSLR camera fitted with a Sigma 35 mm lens was used, which was controlled remotely via a tethered laptop connection.

For track-based LSF, an LED lamp illuminated the specimen obliquely (cf. 45°) or directly for the white light photographs. The long-exposure images under all three lighting regimes for specimens UOP-PAL-SOL-001–013 (figures 2–8; electronic supplementary material, figures S1–S4 and S8–S10) were taken using a Nikon D5300 DSLR camera mounted on a tripod with a 2 s self-timer setting that prevents camera movement from affecting the image.

Aperture priority mode controlled the length of exposure following the method described by Eklund *et al.* [61], who suggested that a 10 s exposure for UV would be sufficient with the ISO being adjusted accordingly. Low ISO values were preferred to minimize grainy aspects on images. 30 s time-exposed images were used for LSF to allow the module to pass across the entire specimen; ISO was set to 100. When using LSF with a green laser, the O56 blocking filter was used to prevent laser over-saturation. Post-processing (equalization, saturation and colour balance) for all images was then performed in Photoshop CS6.

## 2.6. Reproducibility

Thanks to reductions to the cost of laser systems, the track-based LSF method can be replicated using a 532 nm laser, an LCD power supply and a Zecti 31.5in'/80 cm camera slider, as used for specimens UOP-PAL-SOL-001–013 (figures 2–8; electronic supplementary material, figures S1–S4 and S8–10). The success of our set-up, involving an 85 mW laser, is evident from the fact that the technique can be used with moderately powerful laser equipment, compared to the higher power ones used previously (300–500 mW, e.g. [5,11]). The use of a less powerful laser also allows for LSF to be more accessible on the grounds of cost for further studies or as a teaching resource, but with a necessary reduction in fluorescence signal.

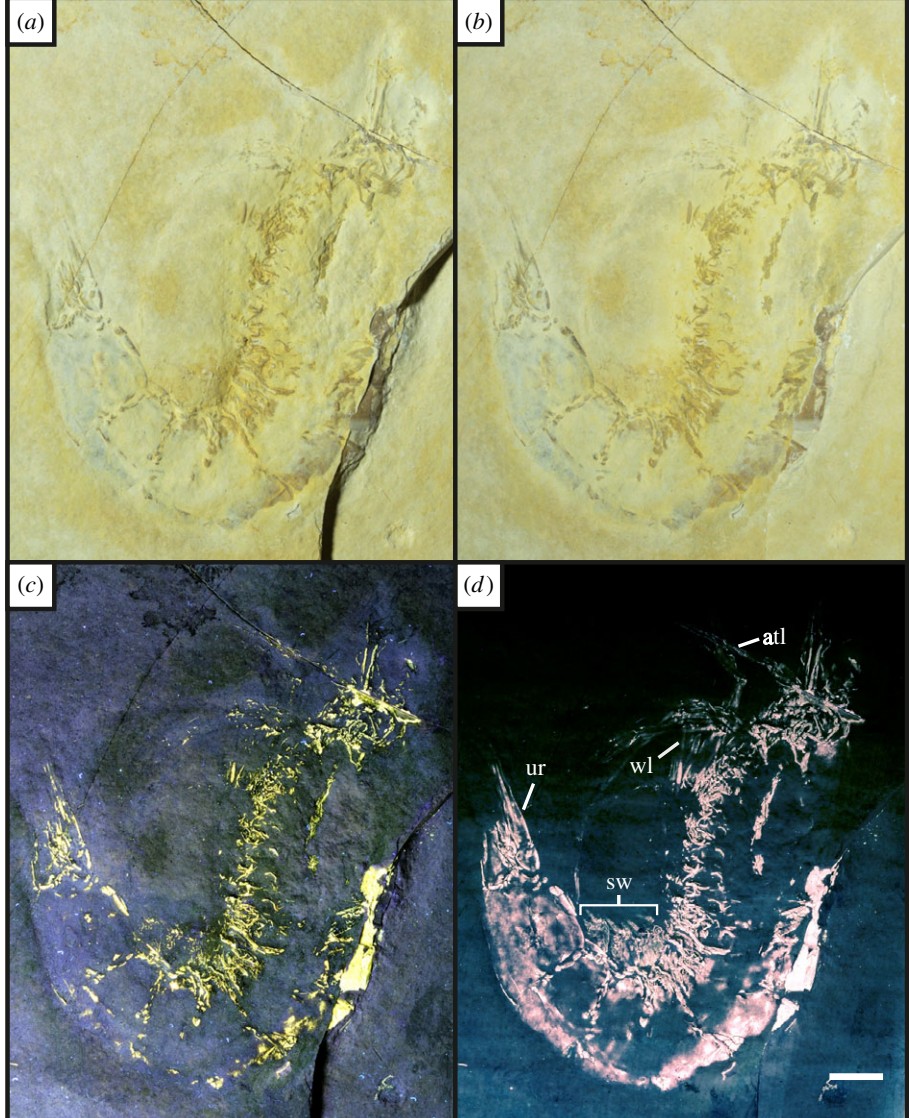

**Figure 5.** Complete fossil of the penaeid shrimp *Antrimpos* sp. [54] (UOP-PAL-SOL−008) under different lighting conditions; (*a*), oblique white light showing the split rock revealing the original exoskeleton; (*b*), direct white light; (*c*), UV-A fluorescence increases the contrast with the background by illuminating the specimen and leaving the matrix dark; (*d*), LSF reveals the entire body outline of the animal, and this not seen in the white light or UV-A photographs. Note the antennulae (atl), swimmerets (sw), walking legs (wl) and urostyle (ur) which were not fully visible in white light. Abbreviations following Haug *et al*. [23]. The green laser wavelength used on this specimen was 532 nm. Scale bar = 10 mm.

## 2.7. Health and safety

UV and LSF are potentially dangerous for the eyes and skin, and precautions must be taken. These methods were conducted in a locked room with a suitable exterior notice to prevent people from being harmed. UV and laser blocking safety goggles were worn during imaging.

## 2.8. Limitations of the study

Track-based LSF was limited to 85 mW power levels due to a lack of high-powered laser equipment within the University of Portsmouth. The UV comparison only used UV-A due to the limited equipment available so future work would benefit from comparisons with UV-B and UV-C. The photos under both UV-A and LSF were sometimes overexposed to enhance the detection of hidden structures. Certain specimens were only imaged with LSF so UV imaging could be compared in future studies e.g. for figure 9 and electronic supplementary material, figures S5–S7. However, the data presented and analysed fully supports the interpretations and conclusions made.

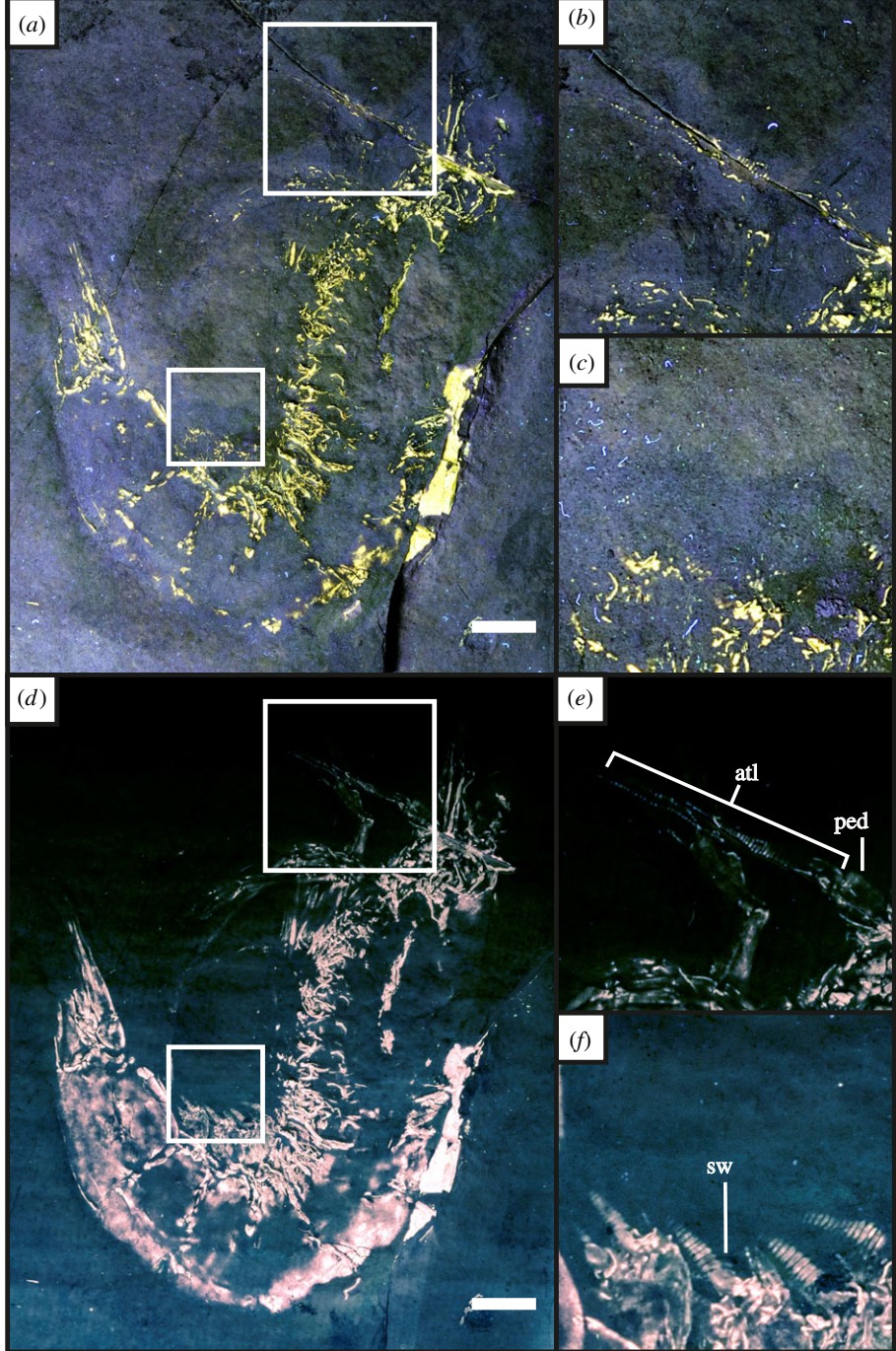

**Figure 6.** Isolated UV-A (*a*–*c*) and LSF (*d*–*f*) images from figure 5 highlighting the differences in revealed detail. (*b*) and (*e*) are UV-A and LSF images of the revealed antennae with clear segmentation visible under LSF. (*c*) and (*f*) highlight the pleopods that are fully revealed under LSF with full segmentation. Abbreviations as above with the addition of the antennular peduncle (ped) from Audo and Charbonnier [56]. Scale bar = 10 mm. (*b*) and (*e*) magnified ×2 and *c* and *f* magnified ×3.6.

## 3. Results

### 3.1. Ultraviolet fluorescence

Under UV-A, soft tissues or original shell material fluoresce yellow-white, contrasting with the blue hue across the rest of the specimen (figures 2–8). UV-A fluoresces material with colour differences across the fossil but with a decreased contrast between non-mineralized elements and the rest of the fossil (e.g. the siphuncle and surrounding shell of *Neochetoceras* sp. [53], figure 2c). Decapods fluoresce readily under UV-A (figures 3–7 [54–57]) which greatly enhances the specimen-matrix contrast. Specimen UOP-PAL-SOL–007

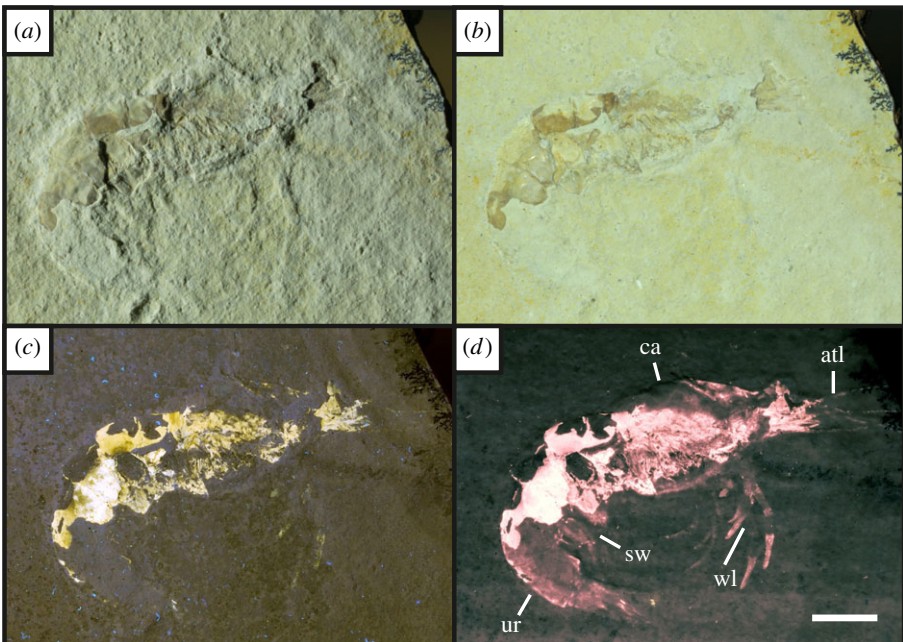

**Figure 7.** Specimen UOP-PAL-SOL–009 of the caridean shrimp *Alcmonacaris* sp. [57] under oblique white light (*a*), direct white light (*b*), UV-A fluorescence (*c*) and LSF (*d*). Notice the faint outline under UV-A is enhanced through LSF to reveal the full animal, allowing the fossil to be labelled. Scale = 10 mm. The features revealed use the same abbreviations as figure 9 with the addition of the carapace (ca). Abbreviations following Haug *et al*. [23]. The LSF wavelength used was 532 nm.

lacks a distal portion of the left first pereiopod, which remains dark under UV-A (figure 4). This juvenile specimen of *Knebelia* sp. [55] is also missing the edge of a telson, which remains dark under UV-A, likely indicating a lack of preserved material resulting in a sole imprint. In *Antrimpos* sp. [54,56] (UOP-PAL-SOL–008), the specimen is largely covered in a thin veneer of the matrix where UV-A appears to illuminate the cuticle exposed at the surface (figures 5*c* and 6*a–c*). This is best observed where the limestone has chipped (bottom right of figure 5*c*), exposing a section of the exoskeleton and as a result, the fluorescence is much brighter. A specimen of *Alcmonacaris* sp. [57] (UOP-PAL-SOL–009) reveals a faint outline of the animal under UV-A while recording differences in preserved cuticle thickness evident through differing levels of fluorescence (figure 7). Fish skeletons fluoresce well under UV-A and the sediment-matrix contrast is increased greatly, although illumination is restricted to material exposed on the surface of the matrix (figure 8*c* [58]; electronic supplementary material, figures S8–S10).

## 3.2. Laser-stimulated fluorescence

Using a green laser for LSF with an O56 blocking filter fluoresces fossils with an orange hue and revealed details with differing degrees of brightness controlled by the amount of original material present. Figure 9 and electronic supplementary material, figures S5–S7 were all studied under a violet laser through a 425 nm longpass filter, resulting in soft tissues, glue and preservation materials fluorescing. Under LSF, the ammonite specimens fluoresce the surrounding shell along with the siphuncle that is present under white light (figure 2*a,b,d*) but was clearer under fluorescence (figure 2*d*; electronic supplementary material, figure S2D). Decapod crustaceans readily fluoresce under LSF, with this fluorescence often extending beyond the preserved exoskeleton, as seen when the technique revealed geochemical halos around the lost calamus of 'Archaeopteryx' [48]. In another study of *Archaeopteryx*, LSF revealed cornified feather sheaths, offering evidence of sequential moulting [49]. Subsurface fluorescence aided in the discovery of missing sections that occur because of incomplete preparation (figures 3–8). The isolated pereiopod of UOP-PAL-SOL–006 (figure 3) under LSF contrasts strongly with the matrix revealing the appendage and its setae in their entirety. The missing section on the distal portion of the left pereiopod of UOP-PAL-SOL–007 is revealed under LSF, again through fluorescence under the thin layer of limestone (figure 4*c,e*). The missing telson remains dark under LSF, and as stated above, is likely an imprint. *Antrimpos* sp. UOP-PAL-SOL–008 (figures 5–6) has not

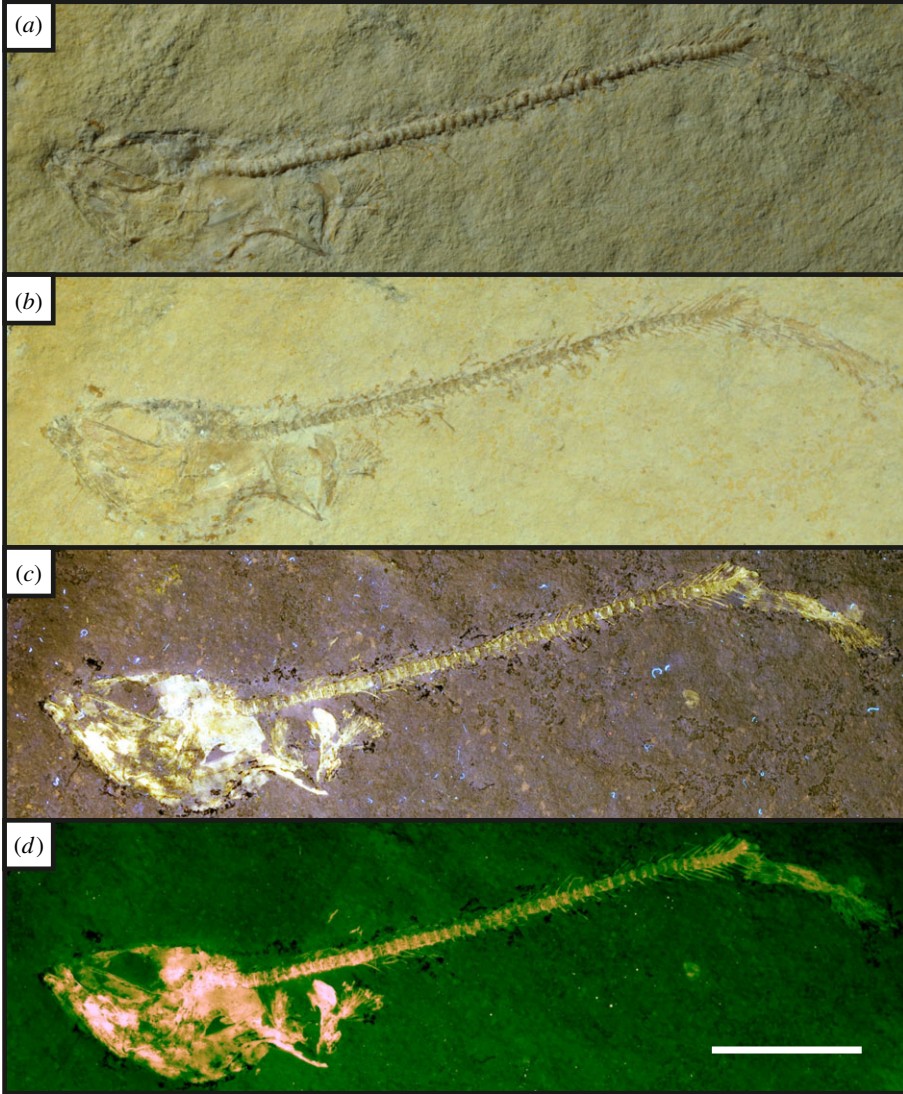

**Figure 8.** A near complete specimen of a fish from the Orthogonikleithridae family [58], lacking the dorsal section of caudal fin (UOP-PAL-SOL–011). (*a*), oblique white light photograph showing a complete vertebral column and faint skull and caudal fin elements; (*b*), direct white light image; (*c*), UV-A fluorescence highlighting the bones of the skull along with the pectoral fin; (*d*), LSF fluorescence of the specimen showing the caudal fin is incomplete along with fluorescence of the entire skull. LSF was carried out with a 532 nm green laser. Scale bar = 20 mm.

been prepared, leaving much of the fossil covered by matrix. Under LSF, a more complete body outline is exposed with a green 532 nm laser (figure 5*d*). Under the matrix, other parts of the animal can be identified (e.g. swimmerets with individual segmentation), often with higher clarity, and this is emphasized in the magnified images of the same specimen (figure 6*e,f*). Under LSF, UOP-PAL-SOL–009 reveals differences in cuticle thickness observed as varying levels of fluorescence along its entirety including in the subsurface (figure 7*d*). One specimen of *Germanodactylus rhamphastinus* [62] studied under tripod-based LSF reveals a distinct contrast between preserved bone and the imprints of the skeleton in the matrix, especially in the skull area (electronic supplementary material, figure S7). Fish fluoresce well under both UV-A and LSF with often minimal difference between the two. The autocentra surrounding the dark chordocentra are thin and almost translucent and fluoresce under both UV-A and LSF. In UOP-PAL-SOL–011 however, LSF has greater contrast than the UV-A photograph, as the higher laser flux and subsurface fluorescence reveals the missing elements of the skull, vertebral column and ventral caudal fin (figure 8*d*). The missing section of the caudal fin is not revealed through LSF or UV-A and is likely missing. The dwarf crocodyliform *Alligatorellus beaumonti* [59] (figure 9) fluoresces under the violet laser and displays soft tissues around the entire body, with particular emphasis on the belly and tail regions. The brighter section

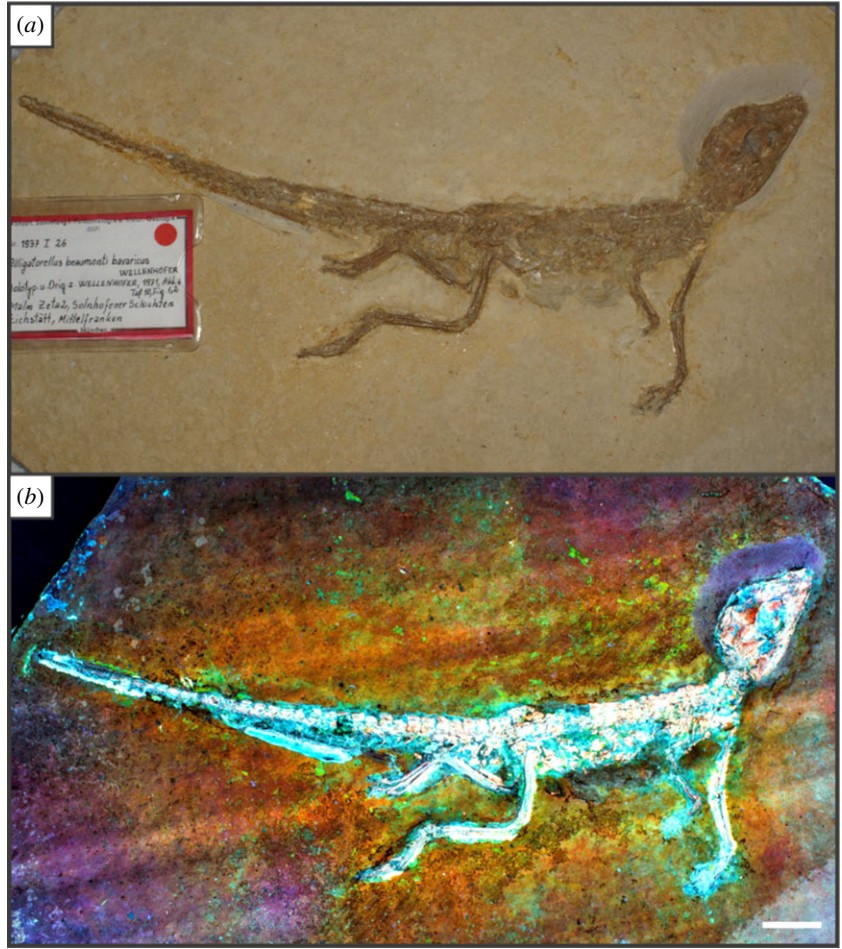

**Figure 9.** Specimen SNSB-BSPG 1937 I 27 of the atoposaurid crocodyliform *Alligatorellus beaumonti* [59] under white light (*a*) and LSF (*b*). The colour patterning makes clear distinction possibly between the skeleton and osteoderms and the surrounding soft tissues. Note the brightly fluorescing soft tissues at the base of the tail, possibly phosphatized remnants of muscle tissue. LSF was carried out using a 405 nm blue laser. Scale = 20 mm.

at the base of the tail may represent partial preservation of the caudofemoralis muscle which is prominent under white light but emphasized through LSF.

## 4. Discussion

The fossils at Solnhofen contrast little with the surrounding sediment, making it difficult to see some fossils without using fluorescence, especially when weathered. The ability to reveal the entire animal, even when the calcified skeleton is covered (see figures 5–7), makes LSF an exciting tool for identifying decapods down to the species level or inferring their ecology [63–65]. This would also reduce preparation time if anatomical information pertinent to distinct taxa can be collected through subsurface detection. Ammonites fluoresce brightly under LSF and UV-A revealing complete shells in some instances, even if it is missing under white light (electronic supplementary material, figure S3). The vertebrate specimens (figures 8 and 9 and electronic supplementary material, figure S5–S10) show exciting results like missing skeletal details under the matrix revealed through subsurface fluorescence (figure 8) and soft tissue preservation (figure 9). The potential caudofemoralis-related soft tissues revealed in figure 9 have important implications for exploring atoposaurid tail function and ecology [66]. The muscle, visible in white light, could be measured for estimated power output and compared to extant and extinct crocodylomorphs. Fish make up the largest proportion of vertebrates in the Solnhofen Limestones with the warm tropical waters home to over 100 species from 70+ genera [52]. Our LSF results could be applied on a larger scale to the fish of the Solnhofen to reveal hidden elements that may be of taxonomic and/or ecological importance [39,67,68]. Our method of UV-A

fluorescence shows good results compared to the white light image as soft-tissues fluoresced, and the specimen-matrix contrast increased. The principal findings in Solnhofen specimens under LSF are that structures revealed under UV-A (e.g. in figures 5–6 where the entire animal is revealed and segmentation is present) are often revealed with greater clarity and subsurface detail, including otherwise unseen details (figures 5–8). Although presented here on macrofossils, laser-based fluorescence has been used on microfossils through confocal laser-scanning microscopy (CLSM) [69]. LSF operates as a simplified form of CLSM with its compact and low-cost set-up [1], and it could be used to study microfossils with ease.

# 5. Conclusion

As mentioned in the introduction, the list of non-destructive techniques available to palaeontologists is growing. Although many have been trialled on Solnhofen fossils to good effect, we present LSF here as an alternative method. Synchrotron Rapid Scanning X-ray Fluorescence (SRS-XRF) produces chemical images that allow for elemental mapping, but this comes with a high cost, complex set-up and restrictions to specimen size. LSF produces a qualitative chemical map of specimens large and small, so it can used as an exploratory aid prior to elemental mapping and other quantitative chemical methods by showing differences in fluorescence signatures that relate to geochemistry.

This study shows the effect of LSF on invertebrates for the first time with decapods providing the best evidence for the use of LSF over UV-A. Ammonites tend to show minimal differences under both techniques, but LSF sometimes fluoresces features more brightly (e.g. in figure 2 where the siphuncle fluoresces more brightly than the surrounding shell). As observed in previous studies too, vertebrates fluoresce well under both UV-A and LSF with the subsurface fluorescence of LSF aiding in detection from within the matrix (e.g. figure 8). This study adds the Solnhofen Plattenkalks to the list of Konservat-Lagerstätten where the effects under LSF are now known, opening fruitful avenues for further studies and the application of LSF to other Mesozoic Plattenkalks such as Cerin, Canjuers, Gara Sbaa and Monte Fallano [70–73]. A major advantage of using LSF in Solnhofen specimens is the subsurface fluorescence of fossils where it can reveal additional information for study that can potentially help to protect tricky specimens that will not be prepared further or help to aid in the preparation process. This study should be considered a first initiative, underscoring the significance of LSF in non-destructive palaeontological investigations with extensive comparative figures that expose unseen details to an equal and often greater extent than UV-A.

# 6. Research availability

## 6.1. Lead contact

Further information and requests for resources and reagents should be directed to and will be fulfilled by the Lead Contact, Luke Barlow (barlowpalaeo@gmail.com).

## 6.2. Materials availability

UOP-PAL-SOL-001–013 are held at the School of the Environment, Geography and Geosciences, University of Portsmouth. Additional specimens from the Staatliche naturwissenschaftliche Sammlungen Bayerns, Bayerische Staatssammlung für Paläontologie und Geologie (SNSB-BSPG) are held in the Bavarian State Collections of Palaeontology and Geology, Munich.

Data accessibility. Original white light, UV-A and LSF photographs have been deposited to Mendeley data and can be found at: https://data.mendeley.com/datasets/ypdrsvygxg/5.
Authors' contributions. Conceptualization: A. B. and L. B.; methodology: A. B., L. B., M. P. and T. G. K.; investigation, L. B., M. P. and T. G. K.; resources, D. M. M., M. P. and T. G. K.; writing—original draft, L. B.; writing—review and editing, L. B., M. P., A. B., D. M. M. and T. G. K.; visualization, L. B.; supervision, A. B., M. P. and L. B.; funding acquisition, M. P. and T. G. K. All authors gave final approval for publication and agreed to be held accountable for the work performed therein.
Competing interests. We declare we have no competing interests.
Funding. M.P. and T.G.K.'s participation in this study was supported by the RAE Improvement Fund of the Faculty of Science, The University of Hong Kong (HKU) and funds from the HKU MOOC Dinosaur Ecosystems. M.P. was also supported by Research Grant Council General Research Fund (grant nos. 17103315; 17120920; 17105221). T.G.K. was

also supported by the Foundation for Scientific Advancement. We would like to thank Oliver Rauhut for granting us access to specimens in his care at the Bavarian State Collections of Palaeontology and Geology.

Acknowledgements. Firstly, we would like to thank our latest reviewers, Ninon Robin in particular along with another anonymous reviewer and the handling editor, Prof. Marcelo Sanchez, for their contributions which have helped us to significantly improve our manuscript. We also thank Sylvain Charbonnier for his thorough and helpful comments on an earlier version of this manuscript. We thank the School of the Environment, Geography and Geosciences at the University of Portsmouth for facilitating this study with special mention to David Loydell for providing the Solnhofen specimens and Richard Hing for accessioning the fossils to the University database.

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
