## [Peer Review File · Royal Society Open Science]

Review History

RSOS-210983.R0 (Original submission)

Review form: Reviewer 1 (Ninon Robin)

Is the manuscript scientifically sound in its present form?

No

Are the interpretations and conclusions justified by the results?

No

Is the language acceptable?

Yes

Do you have any ethical concerns with this paper?

Yes

Have you any concerns about statistical analyses in this paper?

No

Recommendation?

Major revision is needed (please make suggestions in comments)

Comments to the Author(s)

Dear editor (and authors)

This paper represents a simple, concrete and worthwhile study to highlight the important interest of a - not much known - imaging technic adapted to the analysis of flat fossils (a type of preservation that refers to among the most nicely preserved Meso/Cenozoic animals), as in Plattenkalk. At the same time, it provides a new setup context for the applicability/usage of this method "in routine", or at least more commonly, by paleontologists during collection visit or at any lab, without requiring excessive equipment. The illustration set is nicely built and the whole UV/ new lab-based LSF comparison idea is very worth it.

The big weakness of this whole manuscript is its little failure at sticking well to article writing standards, probably explained by the fresh experience of main authors in academia/paleo drafting. This should not be a limit to publication but required though an extensive rework of the manuscript

- partitioning text, per standard article sections
- targeting much better the - to be discussed - real aspects (say less but more specific)
- and very important, re-writing of the results, which, once things have been cleared up, consist in total of only very few sentences per results sub-headings.

In order to do so, I encourage the authors at re-ordering their speech in a much more detailed comparative description, dealing with UV, and then LSF, and this in each of the 3 results sub-heading. I encourage describing much more the particularity of LSF as a rule, per group, and generally (as it is said once well only, in the conclusion), to give much more strength and specificities to the content of the manuscript, that right now is too "volatile". And also to introduce background on the imaging of specific fossils groups somewhere in material. For this, I placed in my review a set of highlights, changes and advices to guide the re-writing, hoping it will be of good help (see Appendix A). Ultimately, a second/last polishing review will have to be brought to the reworked ms, after structural changes.

I finally hope that all authors, of all academic levels, could give an input to this end, to encourage and teach a progression toward the expected standard paper writing.

All best and success with paper modifications,

Review form: Reviewer 2

Is the manuscript scientifically sound in its present form?

No

Are the interpretations and conclusions justified by the results?

Yes

Is the language acceptable?

Yes

Do you have any ethical concerns with this paper?

No

Have you any concerns about statistical analyses in this paper?

No

Recommendation?

Major revision is needed (please make suggestions in comments)

Comments to the Author(s)

Review on RSOS-210983 – Laser-Stimulated Fluorescence reveals unseen details in fossils from the Upper Jurassic Solnhofen Lithographic Limestones of Bavaria, Germany

L. Barlow et al. provide fossil photography using LSF and UV in comparison with “normal light” documentation. This is the first study applying LSF to a variety of macro-fossils from the Bavarian Upper Jurassic plattenkalk lagerstätten (“Solnhofen limestones”). Subsequent to an overview regarding non-destructive imaging methods applicable for such fossils, the authors describe photographic differences on a random series of specimens. In many details LSF shows impressive details, for which a general explanation is performed in a less focused manner.

Abstract:

- The supposed double use for both vertebrate and invertebrate studies deserves no discretion, as long as you do not refer to physical aspects (taphonomy, source of phosphate; see your citation of Wilby et al., 1996, line 67f)
- LSF is said to „surpass... UV methods in some aspects“ – if I’m right in that this advantage is mainly due to subsurface detection, please point out more clearly

General topic and Figures:

- The introduction provides a list of comparable methods. Please explain why the current study was limited to UV vs. LSF? I missed any mentions of Cyan-Red-Photography (-> Haug & Haug, 2011, Fossilien unter langwelligem Licht: Grün-Orange-Fluoreszenz an makroskopischen Objekten. *Archaeopteryx* 29: 20-23.)
- UV is stated to provide “low sensitivity and overemphasising effect on dust” (line 103f). This is a bit poor for a general characterization of UV photography. First, what is meant by sensitivity? The camera’s sensitivity? Or input/radiation, which can be adjusted. Dust, admittedly gleaming under UV, can be removed
- Methods: Once again, UV appears to be less familiar to the authors. The distance, power/ output, duration of exposure and wavelengths must be varied to provide the promised comparison. Some specimens show overexposure, some do not. Virtually nobody successfully working with UV is happy after one certain procedure from a single publication. The entire paragraph (starting line 137) is outstandingly poor, as is the work behind. Only self-optimized procedures on the same specimens can be used for such a comparison, regarding any method included
- Why are the results sorted by taxa? If possible, try to find methodical criteria to structure a methodical study, such as preservation, size of observed section, observed effects, setup, or usability
- Why are some studied fossils not in the “results” section, but in a supplement? Why do some of them miss UV counterparts (including Fig. 8)? Right, it’s said in the limitations section at the end, but why then using them anyway? Please provide a single, well-described result block, even if there’s no UV for all
- Camera setup / Fig. 1: Did the authors use an oblique perspective on the specimen, and if, how was it corrected? Is it possible to integrate a measuring grid, or how? Shouldn’t workers be encouraged to use orthogonal views – maybe this is obvious and also meant in this way, but

I'd recommend to design Fig. 1 according to such settings, to avoid unwanted implications (maybe lift specimen and objective, as the LASER works in an oblique angle anyway)

- Were there any actions to compensate for the grid-like blurring in LSF imaging? Resolution is considerable low under LSF photographing, but as a central part of the results, is not sufficiently addressed in the paper
- The siphuncle of the ammonite (Fig. 2) is visible also under normal light, which is usually found in Solnhofen fossils, so the caption is misleading for readers who are not familiar with such fossils in that the UV and LSF images are not the methods finally revealing this feature, or even details within
- UV isn't UV. The application of UV A, B or C, or combinations, must be tested to compare detailed contrasts, instead of comparing LSF with some generalized "UV" technique. From own experiences, there are "Solnhofen-like" fossils that work only under UV A, while others revealed nothing except under UV C. The conclusions made here do not work at all, as they test only two different procedures, not method vs. method in general
- Readers unfamiliar with the topic will find it easier to read a standardized vocabulary. Please use "UV" or "ultraviolet fluorescence", same with "LSF" vs. "LASER-stimulated fl.", same with the use of A, B, C... (found in some caption's sub-sections)
- In Fig. 3, the use of LSF is convincing as a better tool than UV. Again, no differentiation of UV was performed. Furthermore, the role of intensity (total energy input) was neither controlled nor discussed. Many of the depicted fossils look overexposed (both UV or LSF), which can certainly support the detection of hidden details, but cause heavy blurring on the total image
- For Fig. 4, and others, the detection of missing parts should be discussed in greater detail. Why are some sections missing, or look so? Is that due to locally changed preservation, or incomplete preparation? In the latter case, LASER stimulation might work through thin sheets of limestone – as said in the introduction: subsurface! The authors must underline that the big advantage they found (or applied) is just subsurface detection, not a comparison of various surface methods. In contrast, one edge of the telson is missing (Fig. 4), appearing black in both UV and LSF (due to sole imprint?)
- Identification of the potential caudofemoralis muscle in *Alligatorellus* required more detail, as well as established UV control. The same bluish blurring occurs around the digits or inside the orbit. Would you conclude that this croc had webbed fingers? Or is there indication of matrix rock signals? From my experience, preparation liquids can cause intensive glow under UV – how about LSF then?
- I cannot confirm that (by the comparison in Fig. 9) LSF surpassed UV (line 266) in any way that would concern anatomical documentation
- The discussion is a mixture of introduction stuff, things that are already said, but again not explained (line 300), and awaited outlook. No discussion of true effects is given, which is no surprise due to the lack of any systematic variation of settings. In a methodical paper, I'd expect a table or close meshed structure of advantages/disadvantages, possible ranges, combined values...
- If possible, please give an outlook on possible microscopic use

- Limitations section: This should be part of the methods block (limited equipment, incomplete UV comparison)

Minor comments

- If T. Kaye is a contributing author, is a pers.-comm. citation the right way? (line 45)
- Please correct brackets vs. comma in line 54
- Reference list: Viohl 1998: no short hyphen between limestone and genesis
- I strongly recommend deleting the Archaeopteryx passage (starting line 63) and all related references, as there's no further link to this special field. To underline the meaning of "Solnhofen" plattenkalk fossils, address diversity and disparity, preservation, marine-terrestrial relation, Late Jurassic evolutionary transitions – but not the supposed one iconic textbook missing-link Archaeopteryx which is not even studied in the current submission. – A similar aspect is confusing regarding further citations: please just cite, instead of underlining avian/flight origins (line 93) as a special interest. – Again, the description of vertebrates (starting line 235) should lack a list of just nice fossil groups that were not studied, notably since this is the "results" section
- "Solnhofen" in the used way is a generic term for many and partially not related lagerstaetten (line 110; which is why they are frequently referred to as Solnhofen archipelago, or Solnhofen-like plattenkalk). In the younger literature, this issue is increasingly recognized. Using "Solnhofen Limestones" or any other generalization is an unneeded step back. Same for Jehol: it must be plural (lagerstätten) in line 51
- In the caption of Fig. 9, "D" appears to be missing (typos in capital letters)

The submitted manuscript is partially convincing as a method worth been applied for documentation. Unfortunately, severe issues concerning sharpness and systematic comparison are not addressed, nor documented in detail. Not least the use of UV deserves improvement, at least for a methodical paper

The paper can be published as some sort of pioneer experiment, but must also clearly point out weaknesses (resolution) and so far undescribed comparisons (discretion of UV)

If detection of so far unseen parts (for example, Fig. 6F) bases on incomplete preparation, LSF should be discussed as a chance to protect tricky specimens, or support their final preparation. If the detection of hidden parts is due to chemical remains, instead of subsurface stimulation, please explain in comparison

In total, LSF is far from what I'd call a must-have method for the study of Solnhofen limestones. This can be gained by increased systematic experiments. Until then, the study should be discernable as a first initiative. Nonetheless, major revisions are needed to make this a sound study.

Decision letter (RSOS-210983.R0)

Dear Mr Barlow

The Editors assigned to your paper RSOS-210983 "Laser-Stimulated Fluorescence reveals unseen details in fossils from the Upper Jurassic Solnhofen Lithographic Limestones of Bavaria,

Germany" have made a decision based on their reading of the paper and any comments received from reviewers.

Regrettably, in view of the reports received, the manuscript has been rejected in its current form. However, a new manuscript may be submitted which takes into consideration these comments.

You will see that both reviewers see positive aspects to the paper and believe that the material could be valuable and interesting to readers, however both feel that the standard of presentation is significantly below that required for publication. Making appropriate revision will be quite time-consuming -- not just a case of making limited changes in organisation or limited rewording. That is why the rejection allowing resubmission decision has been made. But Editors and reviewers alike hope that in due course you will be able to resubmit a version of the paper after having considered carefully how to address the general and specific comments of the reviewers.

We invite you to respond to the comments supplied below and prepare a resubmission of your manuscript. Below the referees' and Editors' comments (where applicable) we provide additional requirements. We provide guidance below to help you prepare your revision.

Please note that resubmitting your manuscript does not guarantee eventual acceptance, and we do not generally allow multiple rounds of revision and resubmission, so we urge you to make every effort to fully address all of the comments at this stage. If deemed necessary by the Editors, your manuscript will be sent back to one or more of the original reviewers for assessment. If the original reviewers are not available, we may invite new reviewers.

Please resubmit your revised manuscript and required files (see below) no later than 02-Mar-2022. Note: the ScholarOne system will 'lock' if resubmission is attempted on or after this deadline. If you do not think you will be able to meet this deadline, please contact the editorial office immediately.

Please note article processing charges apply to papers accepted for publication in Royal Society Open Science (<https://royalsocietypublishing.org/rsos/charges>). Charges will also apply to papers transferred to the journal from other Royal Society Publishing journals, as well as papers submitted as part of our collaboration with the Royal Society of Chemistry (<https://royalsocietypublishing.org/rsos/chemistry>). Fee waivers are available but must be requested when you submit your manuscript (<https://royalsocietypublishing.org/rsos/waivers>).

Thank you for submitting your manuscript to Royal Society Open Science and we look forward to receiving your resubmission. If you have any questions at all, please do not hesitate to get in touch.

on behalf of Professor Marcelo Sanchez (Associate Editor) and Peter Haynes (Subject Editor)
openscience@royalsociety.org

Associate Editor Comments to Author (Professor Marcelo Sanchez):
Comments to the Author:

By rewriting the paper and considering the many useful suggestions provided by the reviewers a new submission could be created.

Reviewer comments to Author:

Reviewer: 1

Comments to the Author(s)

Dear editor (and authors)

This paper represents a simple, concrete and worthwhile study to highlight the important interest of a - not much known - imaging technic adapted to the analysis of flat fossils (a type of preservation that refers to among the most nicely preserved Meso/Cenozoic animals), as in Plattenkalk. At the same time, it provides a new setup context for the applicability/usage of this method "in routine", or at least more commonly, by paleontologists during collection visit or at any lab, without requiring excessive equipment. The illustration set is nicely built and the whole UV/ new lab-based LSF comparison idea is very worth it.

The big weakness of this whole manuscript is its little failure at sticking well to article writing standards, probably explained by the fresh experience of main authors in academia/paleo drafting. This should not be a limit to publication but required though an extensive rework of the manuscript

- partitioning text, per standard article sections
- targeting much better the - to be discussed - real aspects (say less but more specific)
- and very important, re-writing of the results, which, once things have been cleared up, consist in total of only very few sentences per results sub-headings.

In order to do so, I encourage the authors at re-ordering their speech in a much more detailed comparative description, dealing with UV, and then LSF, and this in each of the 3 results sub-heading. I encourage describing much more the particularity of LSF as a rule, per group, and generally (as it is said once well only, in the conclusion), to give much more strength and specificities to the content of the manuscript, that right now is too "volatile". And also to introduce background on the imaging of specific fossils groups somewhere in material. For this, I placed in my review a set of highlights, changes and advices to guide the re-writing, hoping it will be of good help. Ultimately, a second/last polishing review will have to be brought to the reworked ms, after structural changes.

I finally hope that all authors, of all academic levels, could give an input to this end, to encourage and teach a progression toward the expected standard paper writing.

All best and success with paper modifications,

Reviewer: 2

Comments to the Author(s)

Review on RSOS-210983 – Laser-Stimulated Fluorescence reveals unseen details in fossils from the Upper Jurassic Solnhofen Lithographic Limestones of Bavaria, Germany

L. Barlow et al. provide fossil photography using LSF and UV in comparison with “normal light” documentation. This is the first study applying LSF to a variety of macro-fossils from the Bavarian Upper Jurassic plattenkalk lagerstätten (“Solnhofen limestones”). Subsequent to an overview regarding non-destructive imaging methods applicable for such fossils, the authors describe photographic differences on a random series of specimens. In many details LSF shows impressive details, for which a general explanation is performed in a less focused manner.

Abstract:

- The supposed double use for both vertebrate and invertebrate studies deserves no discretion, as long as you do not refer to physical aspects (taphonomy, source of phosphate; see your citation of Wilby et al., 1996, line 67f)
- LSF is said to „surpass... UV methods in some aspects“ – if I’m right in that this advantage is mainly due to subsurface detection, please point out more clearly

General topic and Figures:

- The introduction provides a list of comparable methods. Please explain why the current study was limited to UV vs. LSF? I missed any mentions of Cyan-Red-Photography (-> Haug & Haug, 2011, Fossilien unter langwelligem Licht: Grün-Orange-Fluoreszenz an makroskopischen Objekten. *Archaeopteryx* 29: 20-23.)
- UV is stated to provide “low sensitivity and overemphasising effect on dust” (line 103f). This is a bit poor for a general characterization of UV photography. First, what is meant by sensitivity? The camera’s sensitivity? Or input/radiation, which can be adjusted. Dust, admittedly gleaming under UV, can be removed
- Methods: Once again, UV appears to be less familiar to the authors. The distance, power/output, duration of exposure and wavelengths must be varied to provide the promised comparison. Some specimens show overexposure, some do not. Virtually nobody successfully working with UV is happy after one certain procedure from a single publication. The entire paragraph (starting line 137) is outstandingly poor, as is the work behind. Only self-optimized procedures on the same specimens can be used for such a comparison, regarding any method included
- Why are the results sorted by taxa? If possible, try to find methodical criteria to structure a methodical study, such as preservation, size of observed section, observed effects, setup, or usability
- Why are some studied fossils not in the “results” section, but in a supplement? Why do some of them miss UV counterparts (including Fig. 8)? Right, it’s said in the limitations section at the end, but why then using them anyway? Please provide a single, well-described result block, even if there’s no UV for all
- Camera setup / Fig. 1: Did the authors use an oblique perspective on the specimen, and if, how was it corrected? Is it possible to integrate a measuring grid, or how? Shouldn’t workers be encouraged to use orthogonal views – maybe this is obvious and also meant in this way, but I’d recommend to design Fig. 1 according to such settings, to avoid unwanted implications (maybe lift specimen and objective, as the LASER works in an oblique angle anyway)
- Were there any actions to compensate for the grid-like blurring in LSF imaging? Resolution is considerable low under LSF photographing, but as a central part of the results, is not sufficiently addressed in the paper
- The siphuncle of the ammonite (Fig. 2) is visible also under normal light, which is usually found in Solnhofen fossils, so the caption is misleading for readers who are not familiar with such fossils in that the UV and LSF images are not the methods finally revealing this feature, or even details within
- UV isn’t UV. The application of UV A, B or C, or combinations, must be tested to compare detailed contrasts, instead of comparing LSF with some generalized “UV” technique. From own experiences, there are “Solnhofen-like” fossils that work only under UV A, while others revealed

nothing except under UV C. The conclusions made here do not work at all, as they test only two different procedures, not method vs. method in general

- Readers unfamiliar with the topic will find it easier to read a standardized vocabulary. Please use "UV" or "ultraviolet fluorescence", same with "LSF" vs. "LASER-stimulated fl.", same with the use of A, B, C... (found in some caption's sub-sections)

- In Fig. 3, the use of LSF is convincing as a better tool than UV. Again, no differentiation of UV was performed. Furthermore, the role of intensity (total energy input) was neither controlled nor discussed. Many of the depicted fossils look overexposed (both UV or LSF), which can certainly support the detection of hidden details, but cause heavy blurring on the total image

- For Fig. 4, and others, the detection of missing parts should be discussed in greater detail. Why are some sections missing, or look so? Is that due to locally changed preservation, or incomplete preparation? In the latter case, LASER stimulation might work through thin sheets of limestone – as said in the introduction: subsurface! The authors must underline that the big advantage they found (or applied) is just subsurface detection, not a comparison of various surface methods. In contrast, one edge of the telson is missing (Fig. 4), appearing black in both UV and LSF (due to sole imprint?)

- Identification of the potential caudofemoralis muscle in *Alligatorellus* required more detail, as well as established UV control. The same bluish blurring occurs around the digits or inside the orbit. Would you conclude that this croc had webbed fingers? Or is there indication of matrix rock signals? From my experience, preparation liquids can cause intensive glow under UV – how about LSF then?

- I cannot confirm that (by the comparison in Fig. 9) LSF surpassed UV (line 266) in any way that would concern anatomical documentation

- The discussion is a mixture of introduction stuff, things that are already said, but again not explained (line 300), and awaited outlook. No discussion of true effects is given, which is no surprise due to the lack of any systematic variation of settings. In a methodical paper, I'd expect a table or close meshed structure of advantages/disadvantages, possible ranges, combined values...

- If possible, please give an outlook on possible microscopic use

- Limitations section: This should be part of the methods block (limited equipment, incomplete UV comparison)

Minor comments

- If T. Kaye is a contributing author, is a pers.-comm. citation the right way? (line 45)

- Please correct brackets vs. comma in line 54

- Reference list: Viohl 1998: no short hyphen between limestone and genesis

- I strongly recommend deleting the *Archaeopteryx* passage (starting line 63) and all related references, as there's no further link to this special field. To underline the meaning of "Solnhofen" plattenkalk fossils, address diversity and disparity, preservation, marine-terrestrial relation, Late Jurassic evolutionary transitions – but not the supposed one iconic textbook missing-link *Archaeopteryx* which is not even studied in the current submission. – A similar aspect is confusing regarding further citations: please just cite, instead of underlining avian/flight origins (line 93) as a special interest. – Again, the description of vertebrates (starting line 235) should lack a list of just nice fossil groups that were not studied, notably since this is the "results" section

- “Solnhofen” in the used way is a generic term for many and partially not related lagerstaetten (line 110; which is why they are frequently referred to as Solnhofen archipelago, or Solnhofen-like plattenkalk). In the younger literature, this issue is increasingly recognized. Using “Solnhofen Limestones” or any other generalization is an unneeded step back. Same for Jehol: it must be plural (lagerstätten) in line 51
- In the caption of Fig. 9, “D” appears to be missing (typos in capital letters)

The submitted manuscript is partially convincing as a method worth been applied for documentation. Unfortunately, severe issues concerning sharpness and systematic comparison are not addressed, nor documented in detail. Not least the use of UV deserves improvement, at least for a methodical paper

The paper can be published as some sort of pioneer experiment, but must also clearly point out weaknesses (resolution) and so far undescribed comparisons (discretion of UV)

If detection of so far unseen parts (for example, Fig. 6F) bases on incomplete preparation, LSF should be discussed as a chance to protect tricky specimens, or support their final preparation. If the detection of hidden parts is due to chemical remains, instead of subsurface stimulation, please explain in comparison

In total, LSF is far from what I'd call a must-have method for the study of Solnhofen limestones. This can be gained by increased systematic experiments. Until then, the study should be discernable as a first initiative. Nonetheless, major revisions are needed to make this a sound study.

===PREPARING YOUR MANUSCRIPT===

Your revised paper should include the changes requested by the referees and Editors of your manuscript. You should provide two versions of this manuscript and both versions must be provided in an editable format:
 one version identifying all the changes that have been made (for instance, in coloured highlight, in bold text, or tracked changes);
 a 'clean' version of the new manuscript that incorporates the changes made, but does not highlight them. This version will be used for typesetting if your manuscript is accepted.

If you have been asked to revise the written English in your submission as a condition of publication, you must do so, and you are expected to provide evidence that you have received language editing support. The journal would prefer that you use a professional language editing service and provide a certificate of editing, but a signed letter from a colleague who is a native speaker of English is acceptable. Note the journal has arranged a number of discounts for authors

using professional language editing services
(<https://royalsociety.org/journals/authors/benefits/language-editing/>).

===PREPARING YOUR REVISION IN SCHOLARONE===

<https://royalsociety.org/journals/authors/author-guidelines/#supplementary-material> to include a suitable title and informative caption. An example of appropriate titling and captioning may be found at https://figshare.com/articles/Table_S2_from_Is_there_a_trade-

off_between_peak_performance_and_performance_breadth_across_temperatures_for_aerobic_sc
ope_in_teleost_fishes_/3843624.

Author's Response to Decision Letter for (RSOS-210983.R0)

See Appendix B.

RSOS-211601.R0

Review form: Reviewer 2

Is the manuscript scientifically sound in its present form?

Yes

Are the interpretations and conclusions justified by the results?

Yes

Is the language acceptable?

Yes

Do you have any ethical concerns with this paper?

No

Have you any concerns about statistical analyses in this paper?

No

Recommendation?

Accept with minor revision (please list in comments)

Comments to the Author(s)

thanks for your revisions so far - please see text file (Appendix C).

Decision letter (RSOS-211601.R0)

Dear Mr Barlow,

I am pleased to inform you that your manuscript entitled "Laser-Stimulated Fluorescence reveals unseen details in fossils from the Upper Jurassic Solnhofen Limestones" is now accepted for publication in Royal Society Open Science.

on behalf of Professor Marcelo Sanchez (Associate Editor) and Peter Haynes (Subject Editor)
openscience@royalsociety.org

Associate Editor Comments to Author (Professor Marcelo Sanchez):
Associate Editor
Comments to the Author:
Thank you for your careful considerations for the suggestions

Reviewer comments to Author:
Reviewer: 2
Comments to the Author(s)
thanks for your revisions so far - please see text file

Appendix A

Supplementary Figure Captions

Figure S1. Isolated specimen of *Lamellaptychus* (UOP-PAL-SOL-001), a genus for ammonite mouthparts not associated with a specific taxon, under white light (A-B), UV (C) and LSF (D).

A. Direct white light image differentiating between the brown thick spongy layer and the thin crenulated layer beneath; B, oblique white light casting a shadow on the specimen revealing the full outline; C, ultraviolet light of 365 nm wavelength used to display colour differences between the two layers; D, Increased contrast under LSF using a 532 nm green laser allows the full outline of the original shape to be observed. Scale bar = 10 mm.

Figure S2. The oppeliid ammonite *Neochetoceras bous* Oppel, 1862 with a calcified phragmacone, full body chamber outline and preserved aptychi (UOP-PAL-SOL-002). A, oblique white light displaying remnants of the body chamber at the left inside edge; B, UV image displaying the siphuncle, phragmacone and jaw apparatus with colour differences along with possible stomach contents; C, LSF image under a 532 nm green laser with phosphatised siphuncle along with a clear boundary between the phragmacone and body chamber. Scale bar = 10 mm.

Figure S3. The phragmacone of the oppeliid ammonite *Fontanesiella* sp. Arkell *et al.*, 1957 (UOP-PAL-SOL-003) with phosphatised siphuncle under white light (A-B), UV (C), and LSF under 532 nm green laser light (D). Note the colour variation seen under 365 nm UV is not consistent with Fig. 6 ; this may be because of differential preservation. Scale bar = 10 mm.

Figure S4. The plesiototeuthid squid *Plesiototeuthis prisca* (Rüppell, 1829) with a fluorescent gladius (UOP-PAL-SOL-005). This squid can be seen under oblique (A) and direct white light (B). 365 nm UV light fluoresces the rachis in this specimen, showing that the central rachis is raised (C). LSF using the 532 nm laser fluoresces this gladius at different levels with the central vane picked out through its higher fluorescence. Scale bar = 10 mm.

Figure S5. Specimen SNSB-BSPG 1935 I 24 of the pterodactyloid pterosaur *Ctenochasma gracile* Wagner, 1861 under white light (A) and LSF (B). This individual is fully articulated with a blue fluorescent soft tissue body outline and blue cartilage between the light orange bones. A 405 nm blue laser was used to fluoresce this specimen.

Figure S6. Specimen SNSB-BSPG AS I 745a of the pterodactyloid pterosaur *Germanodactylus rhamphastinus* Wagner, 1851 under white light (A) and LSF (B). The skull and upper torso have an increased contrast under LSF through a 405 nm blue laser, separating the specimen from the surrounding matrix. Specimen badge measures 5 cm.

Figure S7. Specimen SNSB-BSPG 1977 XIX 1 of the pterodactyloid pterosaur *Germanodactylus rhamphastinus* on a counterplate slab under white light (A) and LSF (B). The counterplate slab has some missing bone material where it remains dark under LSF. A 405 nm blue laser was used to carry out LSF and fluoresces any preserved bone material.

Figure S8. A specimen of the teleost fish *Leptolepides sprattiformes* Blainville, 1818 (UOP-PAL-SOL-010), under white light (A), UV fluorescence (B) and LSF (C). Note the completed soft tissue outline that is brighter than the ossified material. Gut contents fluoresce brightly at the centre of the body cavity along with a coprolite illuminated at the posterior. The laser used on this specimen was a 532 nm green laser. Scale = 10 mm.

Figure S9. Two small specimens of the salmoniform fish *Orthogonikleithrus hoelli* on the same slab (UOP-PAL-SOL-012a and b, left and right respectively) under different light conditions. A, oblique white light photograph with a dark vertebral column and lighter skull and fins present; B, UV photograph with complete skull and tail revealed. Dorsal and pelvic fins are also revealed through fluorescence with illumination from the left; C, LSF image revealing the entire dorsal and pelvic fins on the right specimen that are faint under UV. Note the erect

position of the dorsal and pelvic fins on the left and relaxed position on the right. A 532 nm green laser was used to carry out LSF. Scale bar = 10 mm.

Figure S10. Specimen UOP-PAL-SOL-013 of the extinct teleost *Tharsis dubius* (Blainville, 1818) under white light (A), UV (B) and LSF (C) conditions. The fluorescence around the articulated skeleton reveals loose scales along with gular plates and gut contents. A 532 nm green laser was used to carry out LSF. Scale bar = 10 mm.

Figure S1. Isolated cephalopod *Lamellaptychus* under white light, UV and LSF.

Figure S2. The oppelliid ammonite *Neochetoceras bous* with a calcified phragmacone, full body chamber outline and preserved aptychi.

Figure S3. The ammonite *Fontannesiella prolithographica* phragmacone under white light (A-B), UV (C), and LSF (D).

Figure S4. Preserved gladius of plesiotheuthid squid *Plesiotheuthis prisca* with body outline in the sediment.

Figure S5. Specimen SNSB-BSPG 1935 I 24 of the pterodactyloid pterosaur *Ctenochasma gracile* under white light (A) and LSF (B).

Figure S6. *Germanodactylus rhamphastinus* under white light (A) and LSF (B). Specimen badge measures 5cm across.

Figure S7. Specimen SNSB-BSPG 1977 XIX 1 of the pterodactyloid pterosaur *Germanodactylus rhamphastinus* on a counterplate under white light (A) and LSF (B).

Figure S8. A specimen of the teleost fish *Leptolepides sprattiformes* under different light conditions.

Figure S9. Two small specimens of the salmoniform fish *Orthogonikleithrus hoelli* Arratia 1997 on the same slab under different light conditions.

Figure S10. Specimen of the extinct bulldog fish *Tharsis dubius* under white light (A), UV (B) and LSF (C) conditions.

Supplementary references – Those looked at but not included in the final manuscript

- Arkell, W., Furnish, W., Kummel, B., Miller, A., Moore, R., Schindewolf, O., Sylvester-Bradley, P., & Wright, C. (1957). Treatise on invertebrate paleontology part L, Mollusca 4. *Geological Society of America and University of Kansas Press, Boulder, 490pp.*
- Blainville, H. de. (1818). Poissons fossiles. Nouvelle Dictionnaire d'Histoire naturelle. *Nouvelle Édition, 27, 334–361.*
- Cuesta, E., Diaz-Martinez, I., Ortega, F., & Sanz, J. L. (2015). Did all theropods have chicken-like feet? First evidence of a non-avian dinosaur podotheca. *Cretaceous Research, 56, 53–59.*
- Hoffmann, R., Bestwick, J., Berndt, G., Berndt, R., Fuchs, D., & Klug, C. (2020). Pterosaurs ate soft-bodied cephalopods (Coleoidea). *Scientific Reports, 10(1), 1–7.*
- Hone, D. W., Tischlinger, H., Xu, X., & Zhang, F. (2010). The extent of the preserved feathers on the four-winged dinosaur *Microraptor gui* under ultraviolet light. *PloS One, 5(2), e9223.*
- Jaeger, K. R., Tischlinger, H., Oleschinski, G., & Sander, P. M. (2018). Goldfuss was right: Soft part preservation in the Late Jurassic pterosaur *Scaphognathus crassirostris* revealed by reflectance transformation imaging and ultraviolet light and the auspicious beginnings of paleo-art. *Palaeontologia Electronica, 21(3).*
- Kellner, A. W., Wang, X., Tischlinger, H., de Almeida Campos, D., Hone, D. W., & Meng, X. (2010). The soft tissue of *Jeholopterus* (Pterosauria, Anurognathidae, Batrachognathinae) and the structure of the pterosaur wing membrane. *Proceedings of the Royal Society B: Biological Sciences, 277(1679), 321–329.*
- Oppel, A. (1862). Über jurassische Cephalopoden. *Palaontologische Mitteilungen Aus Dem Museum Der Koniglich-Bayerischen Staates, 1, 127–266.*
- Racicot, R. (2017). Fossil secrets revealed: X-ray CT scanning and applications in paleontology. *The Paleontological Society Papers, 22, 21–38.*

Simpson, G. G. (1926). Are *Dromatherium* and *Microconodon* Mammals? *Science*, 63(1639), 548–549.

Appendix B

Response to Reviewers

We wish to thank the editor and the two anonymous reviews for their constructive and helpful comments. Please see our point-by-point response below. We hope that you enjoy our revised manuscript.

Editor comments:

Regrettably, in view of the reports received, the manuscript has been rejected in its current form. However, a new manuscript may be submitted which takes into consideration these comments.

You will see that both reviewers see positive aspects to the paper and believe that the material could be valuable and interesting to readers, however both feel that the standard of presentation is significantly below that required for publication. Making appropriate revision will be quite time-consuming - not just a case of making limited changes in organisation or limited re-wording. That is why the rejection allowing resubmission decision has been made. But Editors and reviewers alike hope that in due course you will be able to resubmit a version of the paper after having considered carefully how to address the general and specific comments of the reviewers.

Many thanks for your comments. We believe we have addressed the comments from Reviewer 1 and 2 in our revised manuscript and it has strengthened our manuscript.

Reviewer 1 comments:

This paper represents a simple, concrete, and worthwhile study to highlight the important interest of a - not much known - imaging technic adapted to the analysis of flat fossils (a type of preservation that refers to among the most nicely preserved Meso/Cenozoic animals), as in Plattenkalk. At the same time, it provides a new setup context for the applicability/usage of this method "in routine", or at least more commonly, by paleontologists during collection visit or at any lab, without requiring excessive equipment. The illustration set is nicely built and the whole UV/ new lab-based LSF comparison idea is very worth it.

We thank the reviewer for believing that our manuscript will form a nice submission to this journal as it was our intention to bring this technique to a wider audience along with displaying new comparative results.

The big weakness of this whole manuscript is its little failure at sticking well to article writing standards, probably explained by the fresh experience of main authors in academia/paleo drafting. This should not be a limit to publication but required though an extensive rework of the manuscript

- partitioning text, per standard article sections
- targeting much better the - to be discussed - real aspects (say less but more specific)
- and very important, re-writing of the results, which, once things have been cleared up, consist in total of only very few sentences per results sub-headings.

Many thanks for the constructive bullet points and we have done our best to address the above concerns. This revised manuscript has been reduced in size to take out unnecessary information.

In order to do so, I encourage the authors at re-ordering their speech in a much more detailed comparative description, dealing with UV, and then LSF, and this in each of the 3 results sub-heading. I encourage describing much more the particularity of LSF as a rule, per group, and generally (as it is said once well only, in the conclusion), to give much more strength and specificities to the content of the manuscript, that right now is too "volatile". And also to introduce background on the imaging of specific fossils groups somewhere in material. For this, I placed in my review a set of highlights, changes and advices to guide the re-writing, hoping it will be of good help. Ultimately, a second/last polishing review will have to be brought to the reworked ms, after structural changes.

Thanks so much for providing a detailed set of highlights on the manuscript which became a starting point for the changes made and was extremely helpful to have.

I finally hope that all authors, of all academic levels, could give an input to this end, to encourage and teach a progression toward the expected standard paper writing.

All best and success with paper modifications,

Reviewer 2 comments:

L. Barlow et al. provide fossil photography using LSF and UV in comparison with “normal light” documentation. This is the first study applying LSF to a variety of macro-fossils from the Bavarian Upper Jurassic plattenkalk lagerstätten (“Solnhofen limestones”). Subsequent to an overview regarding non-destructive imaging methods applicable for such fossils, the authors describe photographic differences on a random series of specimens. In many details LSF shows impressive details, for which a general explanation is performed in a less focused manner.

Abstract:

- The supposed double use for both vertebrate and invertebrate studies deserves no discretion, as long as you do not refer to physical aspects (taphonomy, source of phosphate; see your citation of Wilby et al., 1996, line 67f)

- LSF is said to „surpass... UV methods in some aspects“ – if I’m right in that this advantage is mainly due to subsurface detection, please point out more clearly

These points have been noted and changed accordingly making subsurface detection far more pointed out in our revised manuscript.

General topic and Figures:

- The introduction provides a list of comparable methods. Please explain why the current study was limited to UV vs. LSF? I missed any mentions of Cyan-Red-Photography (-> Haug & Haug, 2011, Fossilien unter langwelligem Licht: Grün-Orange-Fluoreszenz an makroskopischen Objekten. *Archaeopteryx* 29: 20-23.)

Well pointed out by the reviewer and cyan-red photography has been mentioned in comparative methods. The study was restricted to UV-A vs LSF as UV-A is well established in palaeontology and the equipment was most accessible. We believe the comparison

between LSF on a new site and other groups of fossils with UV-A is worthwhile.

- UV is stated to provide “low sensitivity and overemphasising effect on dust” (line 103f). This is a bit poor for a general characterization of UV photography. First, what is meant by sensitivity? The camera’s sensitivity? Or input/radiation, which can be adjusted. Dust, admittedly gleaming under UV, can be removed

We believe this has been made clearer now.

- Methods: Once again, UV appears to be less familiar to the authors. The distance, power/output, duration of exposure and wavelengths must be varied to provide the promised comparison. Some specimens show overexposure, some do not. Virtually nobody successfully working with UV is happy after one certain procedure from a single publication. The entire paragraph (starting line 137) is outstandingly poor, as is the work behind. Only self-optimized procedures on the same specimens can be used for such a comparison, regarding any method included

Thank you for the detailed criticism. The blanket studying of fossils under UV-A followed a well-established method so while they could have been further optimised, the figures still show adequate information for a comparison to be made. The paragraph has been reworked to a level we believe is better.

- Why are the results sorted by taxa? If possible, try to find methodical criteria to structure a methodical study, such as preservation, size of observed section, observed effects, setup, or usability

This has been changed and the results now consist of two headings for each technique.

- Why are some studied fossils not in the “results” section, but in a supplement? Why do some of them miss UV counterparts (including Fig. 8)? Right, it’s said in the limitations section at the end, but why then using them anyway? Please provide a single, well-described result block, even if there’s no UV for all

Limitations has now been moved to Materials and Methods section. Results are now in a single block.

- Camera setup / Fig. 1: Did the authors use an oblique perspective on the specimen, and if, how was it corrected? Is it possible to integrate a measuring grid, or how? Shouldn’t workers be encouraged to use orthogonal views – maybe this is obvious and also meant in this way, but I’d recommend to design Fig. 1 according to such settings, to avoid unwanted implications (maybe lift specimen and objective, as the LASER works in an oblique angle anyway)

The figure has been fixed to avoid any confusion, thank you for the suggestions.

- Were there any actions to compensate for the grid-like blurring in LSF imaging? Resolution is considerable low under LSF photographing, but as a central part of the results, is not sufficiently addressed in the paper

No there we no actions, we provide a simple method that others can replicate easily.

- The siphuncle of the ammonite (Fig. 2) is visible also under normal light, which is usually found in Solnhofen fossils, so the caption is misleading for readers who are not familiar with such fossils in that the UV and LSF images are not the methods finally revealing this feature, or even details within

We have now addressed this to prevent misleading future readers. See line 204.

- UV isn't UV. The application of UV A, B or C, or combinations, must be tested to compare detailed contrasts, instead of comparing LSF with some generalized "UV" technique. From own experiences, there are "Solnhofen-like" fossils that work only under UV A, while others revealed nothing except under UV C. The conclusions made here do not work at all, as they test only two different procedures, not method vs. method in general

We disagree that our UV-A method is insufficient for displaying results as it is well documented on in various Solnhofen papers using UV-A (365 nm) which is what was used in this paper. UV-A has been more clearly stated in the paper.

- Readers unfamiliar with the topic will find it easier to read a standardized vocabulary. Please use "UV" or "ultraviolet fluorescence", same with "LSF" vs. "LASER-stimulated fl.", same with the use of A, B, C... (found in some caption's sub-sections)

Thank you for this, we have made these changes.

- In Fig. 3, the use of LSF is convincing as a better tool than UV. Again, no differentiation of UV was performed. Furthermore, the role of intensity (total energy input) was neither controlled nor discussed. Many of the depicted fossils look overexposed (both UV or LSF), which can certainly support the detection of hidden details, but cause heavy blurring on the total image

LSF and UV-A have always been used as tools for detecting unseen details. We agree that the images could be sharper, but the important details are still evident, overexposed or otherwise.

- For Fig. 4, and others, the detection of missing parts should be discussed in greater detail. Why are some sections missing, or look so? Is that due to locally changed preservation, or incomplete preparation? In the latter case, LASER stimulation might work through thin sheets of limestone – as said in the introduction: subsurface! The authors must underline that the big advantage they found (or applied) is just subsurface detection, not a comparison of various surface methods. In contrast, one edge of the telson is missing (Fig. 4), appearing black in both UV and LSF (due to sole imprint?)

Thank you for pointing these out and we agree, making this clearer in our revised manuscript.

- Identification of the potential caudofemoralis muscle in Alligatorellus required more detail, as well as established UV control. The same bluish blurring occurs around the digits or inside the orbit. Would you conclude that this croc had webbed fingers? Or is there indication of

matrix rock signals? From my experience, preparation liquids can cause intensive glow under UV – how about LSF then?

More detail has now been added. Glue has also been mentioned to glow under fluorescence. Starting line 538.

- I cannot confirm that (by the comparison in Fig. 9) LSF surpassed UV (line 266) in any way that would concern anatomical documentation

We disagree with this point as the skull, vertebral column and tail are all different under LSF compared with UV-A.

- The discussion is a mixture of introduction stuff, things that are already said, but again not explained (line 300), and awaited outlook. No discussion of true effects is given, which is no surprise due to the lack of any systematic variation of settings. In a methodical paper, I'd expect a table or close meshed structure of advantages/disadvantages, possible ranges, combined values...

We do not believe a table is necessary, but we have reworked the discussion so that it reads easier.

- If possible, please give an outlook on possible microscopic use

Done, see line 256.

- Limitations section: This should be part of the methods block (limited equipment, incomplete UV comparison)

We agree and it has now been moved.

Minor comments

- If T. Kaye is a contributing author, is a pers.-comm. citation the right way? (line 45)

This has now been removed.

- Please correct brackets vs. comma in line 54

Corrected.

- Reference list: Viohl 1998: no short hyphen between limestone and genesis

This has now been fixed in the revised manuscript.

- I strongly recommend deleting the Archaeopteryx passage (starting line 63) and all related references, as there's no further link to this special field. To underline the meaning of "Solnhofen" plattenkalk fossils, address diversity and disparity, preservation, marine-terrestrial relation, Late Jurassic evolutionary transitions – but not the supposed one iconic textbook missing-link Archaeopteryx which is not even studied in the current submission. – A similar aspect is confusing regarding further citations: please just cite, instead of underlining avian/flight origins (line 93) as a special interest. – Again, the description of

vertebrates (starting line 235) should lack a list of just nice fossil groups that were not studied, notably since this is the “results” section

This has been taken onboard and the sections have been removed in the revised manuscript.

- “Solnhofen” in the used way is a generic term for many and partially not related lagerstaetten (line 110; which is why they are frequently referred to as Solnhofen archipelago, or Solnhofen-like plattenkalk). In the younger literature, this issue is increasingly recognized. Using “Solnhofen Limestones” or any other generalization is an unneeded step back. Same for Jehol: it must be plural (lagerstätten) in line 51

The plural of lagerstätten has now been edited. We do not think the use of Solnhofen limestones is an issue to address.

- In the caption of Fig. 9, “D” appears to be missing (typos in capital letters)

Corrected.

The submitted manuscript is partially convincing as a method worth been applied for documentation. Unfortunately, severe issues concerning sharpness and systematic comparison are not addressed, nor documented in detail. Not least the use of UV deserves improvement, at least for a methodical paper

The paper can be published as some sort of pioneer experiment, but must also clearly point out weaknesses (resolution) and so far undescribed comparisons (discretion of UV)

Thanks for seeing some worth in the paper in its current state and we have done our utmost to make the changes required for resubmission.

If detection of so far unseen parts (for example, Fig. 6F) bases on incomplete preparation, LSF should be discussed as a chance to protect tricky specimens, or support their final preparation. If the detection of hidden parts is due to chemical remains, instead of subsurface stimulation, please explain in comparison

This has been made clearer in the revised manuscript. Line 278.

In total, LSF is far from what I’d call a must-have method for the study of Solnhofen limestones. This can be gained by increased systematic experiments. Until then, the study should be discernable as a first initiative. Nonetheless, major revisions are needed to make this a sound study.

We thank the reviewer for noticing this as a first initiative. We feel that the revisions should be more than sufficient.

Appendix C

Review on **RSOS-211601– Laser-Stimulated Fluorescence reveals unseen details in fossils from the Upper Jurassic Solnhofen Limestones** / revised submission of RSOS-210983

L. Barlow et al. provide fossil photography using LSF and UV in comparison with “normal light” documentation. This is the first study applying LSF to a variety of macro-fossils from the Bavarian Upper Jurassic plattenkalk lagerstätten (“Solnhofen limestones”). Subsequent to an overview regarding non-destructive imaging methods applicable for such fossils, the authors describe photographic differences on a random series of specimens. In many details LSF shows impressive details, for which a general explanation and outlook are performed.

I admit that I’ve overlooked the mention of Cyan-Red-Photography, as the authors also cited Haug & Haug (2011) – my mistake. I’d be interested in a detailed comparison of the usefulness of the methods, but this seems to be a point of a subsequent study, to which I want to encourage the authors.

The methodology regarding UV is still poor and would be better if based on more literature examples and/or experimental experience. It is the key to authenticity that when a new method is initially tested, the conventional standard must be better known than from two citations. I accept that no further images can be produced now, so the discussion (or limitations section) should point out that and how individual variation of the cited UV-A procedure could further optimize photography, resulting in a limited comparability with LSF.

In the first review, I’ve stated that resolution is considerable low under LSF photographing, but as a central part of the results, it is not sufficiently addressed in the paper. This is obvious in the figures and should not be missed in the discussion (found no mentions of “resolution” or “blurring”).

In the first review, I didn’t state that the “UV-A method is insufficient”, as the authors replied. I was just telling that there are “Solnhofen” fossils (from the same localities) that show poor reaction under UV-A, while B or C let them glow in finest details. This is true even if literature examples work under UV-A... I’m quite happy with the discretion of UV spectra in the text.

One of my former comments was: >> In Fig. 3, the use of LSF is convincing as a better tool than UV. Again, no differentiation of UV was performed. Furthermore, the role of intensity (total energy input) was neither controlled nor discussed. Many of the depicted fossils look overexposed (both UV or LSF), which can certainly support the detection of hidden details, but cause heavy blurring on the total image <<

The authors replied: >> LSF and UV-A have always been used as tools for detecting unseen details. We agree that the images could be sharper, but the important details are still evident, overexposed or otherwise. <<

This is why readers will be able to sense that UV as a technique is not truly mastered, just sticking to limited citation. I definitely do not reject the images. But (1st) there are reasons for missing resolution, so please find and discuss them, as a helpful and honest documentation for readers. In case of UV, I do know, but others might not. Regarding LSF, I do not know, and want to learn those reasons from your paper. Please offer future optimizations – give your new vehicle a direction.

And (2nd), what you definitely need to address as a future optimization: resolution and overexpose are indeed relevant for the detection of details. What are “important details”? If light energy, exposure, distance etc. could be optimized and revealed further details, wouldn’t these be important as well, maybe even more? Just

let readers know that there's more to discover (Figs. 3, 4, 7, 8, 9) once these factors are optimized – which should require dozens to hundreds of takes per specimen.

Regarding the fish specimen, I formerly stated: >> I cannot confirm that LSF surpassed UV in any way that would concern anatomical documentation <<

The authors replied: >> We disagree with this point as the skull, vertebral column and tail are all different under LSF compared with UV-A. <<

Here, the authors miss the point: Yes, the UV fish is “all different” from the LSF fish, but the UV image is much sharper, shows the same skeletal details, and would support a documentation much better, even more if not overexposed. And here's the point I cannot answer: Can LSF be truly optimized or not? This would be interesting for the outlook. For the discussion (or results), please don't leave readers puzzling about what's so much better about LSF (in this very image). Here, the advantage is nothing but contrast along the column (bright vs. dark), but exactly the same details as under UV, less sharp.

When it is about an outlook (e.g. regarding microscopic use), I'd expect more than a mention of what has been used. Instead, it's about perspectives, chances and challenges, what has not been done. For example, which optimizations (see above), microscopic tools for macroscopic fossils (laser modules, how to adapt stimulation, camera settings) – this is where readers must be stimulated to go one step further, buy equipment, make own experiences, and then need to cite you on and on!

The formerly made comment on the use of “Solnhofen” is rejected, but not justified in the response (>> We do not think the use of Solnhofen limestones is an issue to address <<). I'm sorry, but this is absolutely unacceptable for a scientific article in palaeontology. The authors provide no provenance for the fossils they deal with.

This issue is not just a local detail, but a matter of geological standards. Not only is there a series of localities along a 100 km E-W distance. The plattenkalk bodies are laterally not connected with each other. They originated in different times and contribute to various, non-contemporary formations, up to stage-level (Kimmeridgian to Tithonian)! Summarizing these plattenkalk fossils as “Solnhofen” is a commercial effect resulting from little interest or knowledge about provenances. As scientists, let us please maintain some basics. The collection labels state a locality, which might be debated or unknown for historical or privately traded specimens, but is not to be ignored.

In the abstract, “Lagerstätte” (singular) is correct, if (and only if) all specimens are from the very Solnhofen locality. If not, it must be “Solnhofen-type/Solnhofen-like Limestones Konservat-Lagerstätten” (plural). I cannot correct all the dozens of further passages. Please follow my original comment.

After this 2nd submission, I'm even more interested in applying LSF. All the more I feel this study was compiled in a rather casual attitude. I started this review wishing to push it through this time, but the revisions are certainly not “more than sufficient”.